# Promotion of Cyst Formation from a Renal Stem Cell Line Using Organ-Specific Extracellular Matrix Gel Format Culture System

**DOI:** 10.3390/gels8050312

**Published:** 2022-05-19

**Authors:** Yusuke Sakai, Yoshihiro Kubo, Nana Shirakigawa, Yoshinori Kawabe, Masamichi Kamihira, Hiroyuki Ijima

**Affiliations:** Department of Chemical Engineering, Faculty of Engineering, Graduate School, Kyushu University, 744 Motooka, Nishi-ku, Fukuoka 819-0395, Japan; yoshi.mt09sp@gmail.com (Y.K.); nana.shirakigawa@kyudai.jp (N.S.); kawabe@chem-eng.kyushu-u.ac.jp (Y.K.); kamihira@chem-eng.kyushu-u.ac.jp (M.K.)

**Keywords:** gel sandwich, extracellular matrix, kidney, renal cyst, maturation, CHK-Q cell

## Abstract

Researchers have long awaited the technology to develop an in vitro kidney model. Here, we establish a rapid fabricating technique for kidney-like tissues (cysts) using a combination of an organ-derived extracellular matrix (ECM) gel format culture system and a renal stem cell line (CHK-Q cells). CHK-Q cells, which are spontaneously immortalized from the renal stem cells of the Chinese hamster, formed renal cyst-like structures in a type-I collagen gel sandwich culture on day 1 of culture. The cysts fused together and expanded while maintaining three-dimensional structures. The expression of genes related to kidney development and maturation was increased compared with that in a traditional monolayer. Under the kidney-derived ECM (K-ECM) gel format culture system, cyst formation and maturation were induced rapidly. Gene expressions involved in cell polarities, especially for important material transporters (typical markers *Slc5a1* and *Kcnj1*), were restored. K-ECM composition was an important trigger for CHK-Q cells to promote kidney-like tissue formation and maturation. We have established a renal cyst model which rapidly expressed mature kidney features via the combination of K-ECM gel format culture system and CHK-Q cells.

## 1. Introduction

Polar epithelial cells are a cultured cell model useful to evaluate material transportation for drug screening [1,2,3]. Additionally, many have studies reported that reconstructed in vitro tissue models, such as the kidney and liver, are promising for drug transport and metabolic testing [4,5,6,7]. Kidney-like tissue can be generated by differentiating stem cells based on embryology. For example, differentiation induction of human-induced pluripotent stem cells (iPSCs) for 21–35 days formed kidney organoids [8]. Nephron organoids were formed in 3–4 weeks from human embryonic stem cells (ESCs) and iPSCs [9]. These organoids represent an advanced kidney model, but a combination of adding growth factors and cytokines during induction steps extends the culture period.

Scaffold design is one of the key parts in accelerating cell differentiation and kidney-like tissue formation. Rat renal stem/progenitor cells cultured in Matrigel simultaneously with the addition of growth factors formed kidney-like tissues in 2 weeks [10]. Human primary renal cell aggregation promoted kidney-like cystic tissue formation after 10 days in collagen gel [11]. Although the cysts lack the complex structure of the kidney, they can be used as a simple kidney-like tissue model. Furthermore, the short culture period and simple preparation procedure are attractive for application tools such as material transportation tests. The organ-specific extracellular matrix (ECM) regulates the surrounding environment, affecting development and functional expression [12,13,14]. Human ESCs differentiate toward the kidney when cultured on ECM derived from kidney or lung [15]. Thus, the scaffold environment, similar to the organ from which cells are derived, is important factors affecting the phenotype of cultured cells for kidney-like tissue formation.

Together, the combination of cell types and culture systems with special substrates are important considerations for the rapid fabrication of a kidney-like tissue. Per the existing literature, the long-term supply of multiple factors and/or feeder cells are required for the passage culture and differentiation of ESCs and renal progenitor cells [16,17]. This complex culture process increases culture costs and often hinders applications. Based on the above-mentioned issues, we focused on CHK-Q cells, which are spontaneously immortalized from the renal stem cells of the Chinese hamster [18]. CHK-Q cells grow actively without special culture medium and feeder cells. The growth limitation of CHK-Q cells can induce functional expression. Therefore, CHK-Q cells are a promising cell source for kidney-like tissue construction, although further research is needed for its widespread use.

We developed an excellent culture system for rapidly forming a kidney-like tissue model using CHK-Q cells. We focused on the organ-derived ECM as a trigger to promote cell organization. In particular, the cell–cell contact environment and a three-dimensional (3D) gel format culture system using kidney-derived ECM (K-ECM), which accelerates cell aggregation and the construction of matured kidney-like tissues.

## 2. Results and Discussion

### 2.1. Results

#### 2.1.1. Promotion of Renal Cyst Formation in Collagen Gel Sandwich Culture

The rapid proliferation of CHK-Q cells as a monolayer was observed on a collagen-coated dish, under collagen gel after adhering to a collagen-coated dish, and on collagen gel culture (Figure 1A). In contrast, CHK-Q cells in a collagen gel culture formed 3D cyst-like tissues. More cysts were formed in a gel sandwich culture compared with a gel-embedding culture where the cells were dispersed (shown with * in Figure 1A).

Time-lapse observation revealed cyst formation and fusion history (Figure 1B). On the first day of culture, cells began to gather and 20 μm cysts were formed. Each cyst expanded until day 3. Then, the cyst walls were reconstructed (yellow arrowhead), and the cysts were grown by fusion. On day 7 of the culture, 3D observation of the cytoskeleton confirmed cyst fusion, maintaining the 3D cyst structure (Figure 1C and Appendix A).

#### 2.1.2. Differentiation of Cysts of CHK-Q Cells to Mature Kidney-Like Cells

Gene expression profiles for mesoderm, ureteric bud, collecting duct, nephron progenitor cells and metanephric mesenchyme, renal corpuscle, and uriniferous tubule-related genes are shown as a heatmap (Figure 2). During cyst formation under the collagen gel sandwich culture, there were numerous highly expressed genes associated with kidney development, such as the mesoderm, ureteric bud, and nephron progenitor cell and metanephric mesenchyme, on day 3 of culture (Figure 2A–C). In particular, the cyst tissues showed higher levels of gene expression involved in transcription factor, Wnt signaling, and growth factor for kidney maturation compared with CHK-Q cell suspensions and monolayers (collagen-coated dish). The nephron progenitor marker was upregulated on later culture days. Kidney differentiation markers such as uriniferous tubules and collecting ducts were increased over the conventional 2D monolayer (Figure 2E,F). Genes related to transporters such as Aqp1 were notably upregulated.

#### 2.1.3. Effect on Cyst Formation by Seeding Density or Organ-Derived ECM

CHK-Q cells formed cysts by latest on day 3 of culture in the collagen sandwich culture, regardless of seeding density (Figure 3A). Cysts fused and expanded by culture day 7. Cyst fusion reduced the number of cysts (Figure 3B) and increased their area (Figure 3C). The highest seeding density demonstrated large cysts from the beginning. The confluency increased with the seeding density (Figure 3D). Additionally, the 3D thickness increased with the seeding (Figure 3E).

Under the K-ECM gel format culture system, cyst formation, rapid fusion, and expansion were observed compared with the collagen gel (Figure 4A–C). In L-ECM, small cysts were formed, and fusion was restricted. Culture confluency on day 7 was similar regardless of the gel components (Figure 4D). Then, an increase in 3D thickness was observed within the collagen gel condition compared with an organ-derived ECM gel culture system (Figure 4E).

#### 2.1.4. Gene Expression Changes Related to Kidney Development by Kidney-Derived ECM

The expression of typical markers for the mesoderm, nephron progenitor, and mature renal cells was investigated using real-time RT-PCR. The CHK-Q cells in the collagen gels tended to express increased mesoderm (T) and renal progenitor cell markers (Osr1, Pax8) at lower seeding densities (Figure 5A–C); however, significant differences were observed only on culture day 7 (Figure 5B). The differences in mature kidney markers such as uriniferous tubules and collecting ducts were minimal (Figure 5D–H). Conversely, K-ECM gel format culture system strongly promoted the expression of mature kidney markers, such as transporter (*Slc5a1*), potassium channel (*Kcnj1*), and transcription factor (*Gata3*) (Figure 5E–G). In the L-ECM condition, these markers were less expressed or downregulated compared with the collagen gel conditions (Figure 5D–H).

### 2.2. Discussion

Polar epithelial cells have been used as an organ model for membrane transport tests of drugs [1,2,3,4,5,6,7]. Caco-2 cells derived from human colorectal adenocarcinoma are most commonly used for such tests. They form polar differentiated intestinal epithelium-like tissues after 3 weeks of over-confluent culture [1,19]. MDCK cells, an epithelial cell line derived from canine kidney, can be used after just a few days of culture [20]. MDCK cells autonomously form a domed liquid pool between the substrate and cell monolayer. After a week, the lumen forms under the collagen gel.

Creating a stem cell-based kidney model can be an attractive alternative tool. Supplementing cytokines and organ-specific factors based on embryology greatly contributes to stem cell maturation and the stable expression of organ-specific functions [8,9,10]. A 3D culture also promotes cell differentiation. The formed cellular organoids enable the reconstruction of 3D structures that mimic the living body and promote the restoration of cell polarity [21]. However, in either method, no combination of cell type and culture system immediately converts to functional differentiation with cell polarity.

Generally, cell proliferation and functional expression are inversely correlated. CHK-Q cells can switch from cell growth to protein production via the addition of small molecular compounds (Kawabe, Y., et al., unpublished data). This feature is positive for the development of functional kidney models. However, it is unclear how the scaffold dependence alters cell phenotype. Here, the characteristics of CHK-Q cells were investigated when they were cultured in organ-derived ECM gel format culture systems.

CHK-Q cells grown as a monolayer on collagen-coated dishes and collagen gel grew actively with a doubling time of approximately 15 h. CHK-Q cells did not show 3D morphology like MDCK cells even after reaching confluence (Figure 1A). In contrast, cystic tissues were formed immediately in the gel sandwich culture system (Figure 1B). Cell growth was dramatically restricted by both 3D scaffold-embedding and cell–cell adhesion conditions, and CHK-Q cells were induced to functional differentiation. Gene expression of Mki67, a cell growth marker, decreased to 1/80 (DNA microarray result). Cyst formation was more prominent in collagen gel sandwich culture, which is abundant in cell–cell adhesions compared with gel-embedded culture. Thus, for CHK-Q cells, the gel sandwich culture system may be the only trigger to switch from cell growth to functional differentiation. This feature of CHK-Q cells is important in rapid preparing a tissue model.

CHK-Q cells in the collagen gel sandwich culture formed cysts after 1 day of culture (Figure 1B). Maturation to kidney-like tissue was confirmed via microarray analysis on day 3 of culture (Figure 2). Because the cysts were characterized by both ureter bud and metanephric mesenchyme, cross-interactions and subsequent inductions to both differentiations will promote kidney maturation [22]. Consequently, it was hypothesized that kidney differentiation markers related to uriniferous tubules and collecting ducts were strongly expressed. Data analysis showed that gene expressions at all parts of uriniferous tubules were enhanced (Appendix A). Gene group expressions related to proximal tubules and distal convoluted tubules were particularly increased.

The differentiation from stem cell to kidney-like cell is processed by adding various growth factors and/or through the formation of cellular organoids over several weeks [23,24,25]. For example, it takes 2–4 weeks using optimized methods for kidney-like tissue formation from human iPSCs or rat renal progenitor cells [8,9,10]. Even when human primary renal cells were used, cysts were formed after 10 days in collagen gel-embedded culture [11]. In contrast, CHK-Q cells capable of cyst formation using the gel format culture system showed accelerated maturation. This process requires no growth factor addition and is relatively fast. This methodology reduces culture medium consumption and costs. It will be also useful for drug discovery assays.

The high cell density of CHK-Q cells increased the cell–cell contact. It affected the number and size of the cysts, but the differentiation to the kidney was not so different (Figure 3 and Figure 5). This result did not support the theory that the organoid size of stem cells determines the direction of differentiation [26,27]. In CHK-Q cell culture, cyst formation might work as a special strong trigger for kidney component cell maturation.

Although human ESCs promoted renal differentiation when cultured on lung- and kidney-derived ECM, no difference was observed with organs of origin [15]. In contrast, organ-derived ECM contributed profoundly to the maturation of CHK-Q cells into the kidney-like cells. In K-ECM conditions, the discriminative gene expression patterns showed high expression of mature kidney markers from metanephric mesenchymes, such as proximal tubules and intermediate tubules (Figure 5E,F). Kidney ECM contains type-IV collagen, type-I collagen, laminin, and sugar chains, in that order [28]. Reports suggest that the expression of type-IV collagen and laminin is important for the formation of kidney based on embryology [29]. A portion or combination might be mimicked in the K-ECM condition. However, the gene expression pattern when using L-ECM was not so different from that of the type-I collagen gel. This observation was supported by the fact that the main component of L-ECM is type-I collagen [30]. In the CHK-Q cell culture, the direction of differentiation depends on the organ-derived scaffold. In the future, based on the K-ECM component, it is necessary to clarify how the combination of ECM components affects cyst formation and kidney maturation from CHK-Q cells.

## 3. Conclusions

Here, rapid cyst formation and kidney maturation from CHK-Q cells were achieved using the novel organ-derived ECM gel culture system. CHK-Q cells in a collagen gel sandwich were promoted to form cysts early, on the first culture day. Maturation toward kidney-like tissue can be due to cyst fusion and enlargement. Under the K-ECM gel format culture system, cyst formation and maturation were rapidly achieved. In particular, it was found that the cell polarity of transporters, which are important for material transport, was restored. In this model, the expression of mature kidney features was rapidly enhanced by the combination of K-ECM and CHK-Q cells. In addition, the culture system does not require growth factors for the maintenance and maturation processes of CHK-Q cells. These advantages lead to the reduction of the culture medium consumption and cost and could be important for assay applications. In the future, practical functions such as protein expression and drug transport verification will be required.

## 4. Materials and Methods

### 4.1. Subculture of CHK-Q Cells

CHK-Q cells, which are spontaneously immortalized from the renal stem cells of the Chinese hamster, were kindly provided by Prof. Kamihira [18]. To maintain the characteristics of CHK-Q cells, the cells, cryopreserved in liquid nitrogen using a preservation medium (Cellbanker 2; Takara Bio, Shiga, Japan), were used for each independent experiment. CHK-Q cells were cultured in 90 mm diameter collagen-coated dishes (Asahi Techno Glass, Tokyo, Japan) at 1.8 × 10^5^ cells/cm^2^ with 10 mL of D-MEM/Ham’s F-12 (Fujifilm Wako Pure Chemical, Osaka, Japan), supplemented with 10% fetal bovine serum (Biowest, Nuaillé, France), 100 U/mL penicillin, and 100 mg/mL streptomycin (Thermo Fisher Scientific Inc., Waltham, MA, USA) as a continuous monolayer. After 2 days of culture, cells reaching 90% confluence were treated with a 0.25% trypsin-EDTA solution (Invitrogen) and subcultured at 5.3 × 10^4^ cells/cm^2^. Cells at passage 2 were used for the experiments.

### 4.2. Preparation of Organ-Derived ECM

Healthy adult pig kidney and liver were purchased from Fukuokashokunikuhanbai Co., Ltd. (Fukuoka, Japan). Blood was removed from organs via perfusion of calcium- and magnesium-free phosphate-buffered saline (CMF-PBS) containing 5 mM ethylene glycol-bis(2-aminoethylether)-N,N,N′,N′-tetraacetic acid (EGTA) (Dojindo Laboratories, Kumamoto, Japan), and cryopreserved at −80 °C until use.

Organ-derived ECM was prepared according to the literature [31]. Briefly, organs were cut into about 2 mm × 2 mm × 2 mm pieces using a scalpel. Tissues were soaked in CMF-PBS containing 1% Triton X-100 (Fujifilm Wako Pure Chemical, Osaka, Japan) for 4 days and washed with CMF-PBS for 4 days. These solutions were changed daily and stirred slowly at 4 °C. Dialysis was performed using the Spectra/Por 6 dialysis membrane (MCWO: 1000, Spectrum Laboratories, Inc., Milpitas, CA, USA) at 4 °C for 2 days to remove salts and impurities. Decellularized organs were lyophilized for 24 h, milled, solubilized using 1 mg/mL pepsin (Sigma-Aldrich, St. Louis, MO, USA) in 0.01 N HCl at room temperature for 2 days, and stored at 4 °C until use. The concentrations of organ-derived ECM solutions were calculated from the amount of added components and insoluble matter.

### 4.3. Gel Sandwich Culture

Type-I collagen (Cellmatrix Type I-A; Nitta Gelatin, Osaka, Japan), K-ECM, and liver-derived ECM (L-ECM) solutions were mixed as shown in Appendix A. Mixtures were added to a culture plate at 52 μL/cm^2^ and incubated at 37 °C for 1 h to form a gel. CHK-Q cells were inoculated at 0.33, 1.0, 1.7, 2.3, and 3.5 × 10^5^ cells/cm^2^ and adhered for 1 h. Mixtures were added to the cells. After gel sandwich formation, culture medium was added and changed every 2 days.

Several culture conditions using type-I collagen gel were performed to compare the characters in CHK-Q cells. Type-I collagen mixture was added to CHQ-cells after they adhered to a collagen-coated dish (under collagen gel). CHK-Q cells were also inoculated on a collagen gel substrate (on collagen gel). To evaluate the gel-embedding condition, CHK-Q cells were cultured in a collagen gel with uniform cell dispersion (collagen gel-embedding). Under all the above-mentioned conditions, the CHK-Q cells were inoculated at 1.0 × 10^5^ cells/cm^2^.

### 4.4. Cyst Formation Assay

To investigate CHK-Q cell morphology under several culture conditions, CHK-Q cells were cultured on collagen-coated dishes (Asahi Techno Glass, Tokyo, Japan), under collagen gel, on collagen gel, in a collagen gel sandwich, and embedded in the gel at 1.0 × 10^5^ cells/cm^2^. To track cyst formation, CHK-Q cells in a collagen gel sandwich (1.7 × 10^5^ cells/cm^2^) were time-lapse observed using a phase-contrast microscope (BZ-9000; Keyence Corporation, Osaka, Japan).

CHK-Q cells in a collagen or organ-derived ECM gel sandwich were fixed with 10% formalin (PFA) in CMF-PBS (Fujifilm Wako Pure Chemical, Osaka, Japan) for 30 min. Fixed samples were permeabilized with 0.1% Triton X (Sigma) in CMF-PBS for 30 min and blocked in 1% bovine serum albumin (BSA) for 1 h. They were incubated with CMF-PBS buffer containing 5 unit/mL rhodamine phalloidin (Thermo Fisher Scientific Inc., Waltham, MA, USA), 2 μg/mL Hoechst 33342 (Dojindo Laboratories, Kumamoto, Japan), and 1% BSA for 1 h. Fluorescence images were captured using a fluorescence microscope or a confocal laser scanning microscope (TCS SP8; Leica Microsystems, Wetzlar, Germany). To characterize cyst morphology, the number and area of cysts were analyzed from fluorescence images using ImageJ software version 1.53e. Cyst thickness was analyzed from 3D confocal images.

### 4.5. DNA Microarray Analysis

CHK-Q cell suspension, monolayer on collagen-coated dish (2.0 × 10^3^ inoculated-cells/cm^2^, day 3), and collagen gel sandwich (1.7 × 10^5^ inoculated-cells/cm^2^, day 3 and day 7) were used for total RNA extraction using TRIzol. DNaseI removed genomic DNA contamination (Qiagen, Hilden, Germany). Total RNA quality was measured using an Agilent 2200 TapeStation (Agilent Technologies, Santa Clara, CA, USA). DNA microarray analysis for Cricetulus griseus was conducted using commercially available DNA chips (cat. no. G4858A#077089, single color 8 × 60 K; Agilent Technologies, Santa Clara, CA, USA). Microarray analysis was processed in Cell Innovator Co., Ltd. (Fukuoka, Japan).

Gene markers related to kidney differentiation were analyzed [32]. Gene expression heat map generation and hierarchical clustering of mesoderm, ureteric bud, collecting duct, nephron progenitor cells and metanephric mesenchyme, renal corpuscle, and uriniferous tubule-related genes were conducted using Heatmapper (http://www.heatmapper.ca/; 3 June 2021). Pearson’s correlation was selected as the distance metric; average linkage clustering was selected as the linkage method.

### 4.6. Gene Expression Analysis (Real-Time Polymerase Chain Reaction)

Total RNA of CHK-Q cell suspension, monolayer on collagen-coated dish (2.0 × 10^3^ inoculated-cells/cm^2^, day 3), collagen gel sandwich (0.33, 1.7, and 3.5 × 10^5^ inoculated-cells/cm^2^, day 3 and day 7), and organ-derived ECM gel sandwich (1.7 × 10^5^ inoculated-cells/cm^2^, day 3 and day 7) were extracted using spin columns (NucleoSpin RNA; Macherey-Nagel, Düren, Germany) according to manufacturer’s instructions. cDNA was synthesized from total RNA using a high-capacity cDNA reverse transcription kit (Thermo Fisher Scientific Inc., Waltham, MA, USA). Samples were stored at −30 °C until PCR processing.

The QuantStudio™3 Real-Time PCR System and TaqMan Gene Expression Assay Kit (Thermo Fisher Scientific Inc., Waltham, MA, USA; Appendix A) were used for PCR. The reaction mixture contained 1 μL cDNA sample, 5 μL TaqMan Fast Advanced Master Mix solution, and 4 μL nuclease-free water in a plate predispensed with the TaqMan Gene Expression Assay probes (Thermo Fisher Scientific Inc., Waltham, MA, USA). Forty amplification cycles consisted of 1 s at 95 °C and 20 s at 60 °C. The comparative cycle time (ΔΔCT) method was used to quantify gene expression levels. Expression levels were normalized to *Gapdh*, and CHK-Q cell suspension was the control condition.

### 4.7. Statistical Analysis

Data are presented as mean ± standard deviation (SD). Means of continuous numerical variables were compared via one-way or two-way analyses of variance (ANOVA) on GraphPad Prism version 9.1.2 for Windows (GraphPad Software Inc, San Diego, CA, USA). Values of ** *p* < 0.01 and * *p* < 0.05 were considered statistically significant compared to several conditions within the same experimental time (Tukey’s multiple comparison test); †† *p* < 0.01 and † *p* < 0.05 were statistically significant compared within the culture time under the same conditions.

## Figures and Tables

**Figure 1 gels-08-00312-f001:**
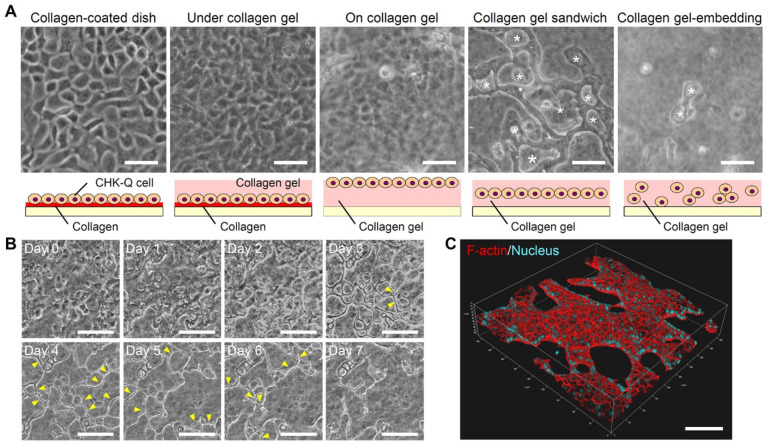
CHK-Q cell morphologies. (**A**) Phase-contrast micrographs at 1.0 × 10^5^ cells/cm^2^ after 3 culture days. Asterisk: cyst tissues. (**B**) Time-lapse observation and (**C**) 3D image of a cyst with collagen gel sandwich at 1.7 × 10^5^ cells/cm^2^. Yellow arrowhead: reconstructed cyst wall. Red (rhodamine phalloidin): F-actin, Blue (Hoechst 33342): Nuclei. Bars represent 100 μm.

**Figure 2 gels-08-00312-f002:**
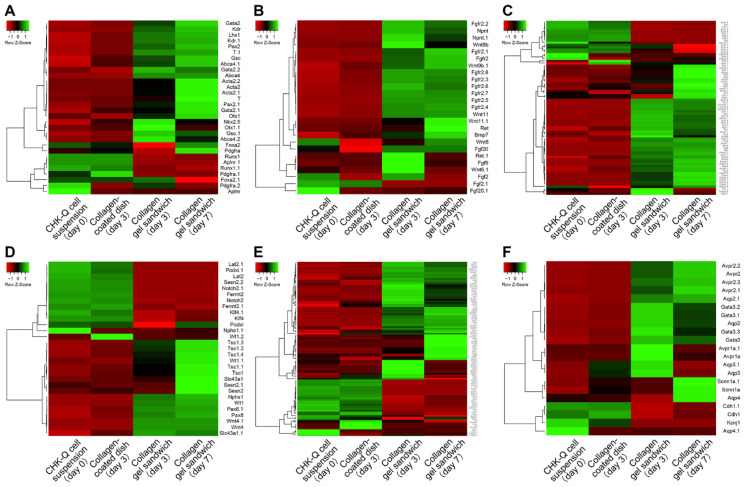
Gene expression heat map of (**A**) mesoderm, (**B**) ureteric bud, (**C**) nephron progenitor cells and metanephric mesenchyme, (**D**) renal corpuscle, (**E**) uriniferous tubule, and (**F**) collecting duct-related genes. Pearson’s correlation was selected as the distance metric; average linkage clustering was selected as the linkage method.

**Figure 3 gels-08-00312-f003:**
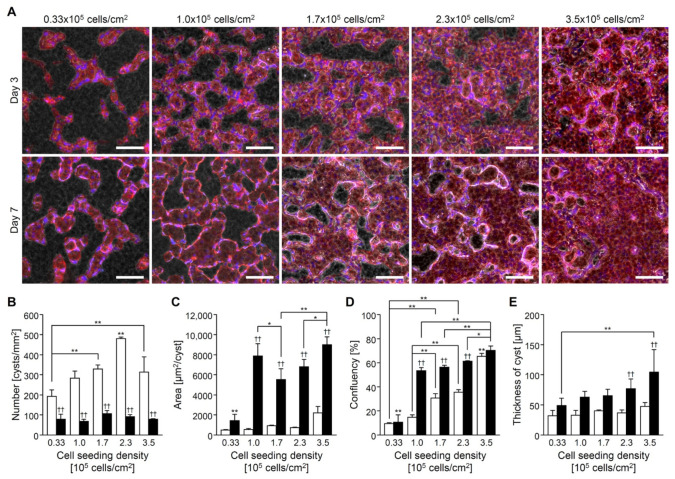
CHK-Q cell cyst characteristics with collagen gel sandwich culture system at several cell density conditions. (**A**) Merged images of phase-contrast and fluorescent micrographs. Red (rhodamine phalloidin): F-actin. Bars represent 100 μm. (**B**) Number, (**C**) area, (**D**) confluence, and (**E**) cyst thickness (*n* ≥ 3). Open and closed columns represent 3 and 7 days of culture, respectively. Data presented as the mean ± SD, ** *p* < 0.01 and * *p* < 0.05 (compared with culture conditions within the same culture time), and †† *p* < 0.01 (compared within the culture time under the same culture conditions) (two-way analyses of variance).

**Figure 4 gels-08-00312-f004:**
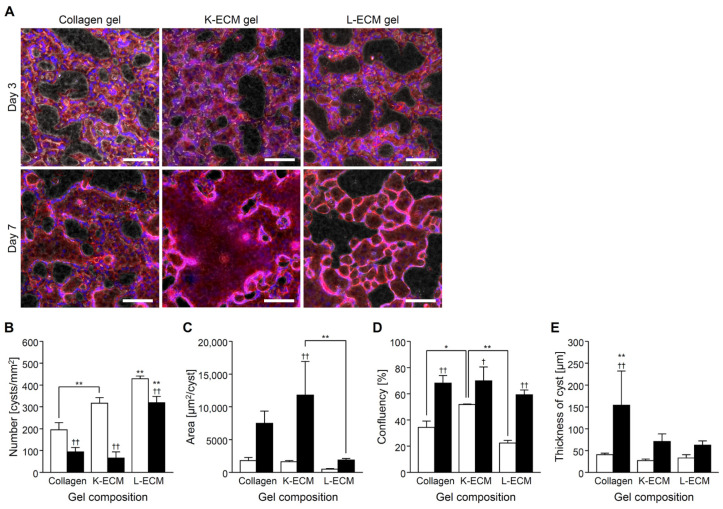
CHK-Q cell cyst characteristics with gel sandwich culture system of several extracellular matrix components at 1.7 × 10^5^ cells/cm^2^. (**A**) Merged images of phase-contrast and fluorescent micrographs. Red (rhodamine phalloidin): F-actin. Bars represent 100 μm. (**B**) Number, (**C**) area, (**D**) confluence, and (**E**) cyst thickness (*n* ≥ 3). Open and closed columns represent 3 and 7 days of culture, respectively. Data presented as the mean ± SD, ** *p* < 0.01 and * *p* < 0.05 (compared with culture conditions within the same culture time), †† *p* < 0.01 and † *p* < 0.05 (compared within the culture time under the same culture conditions) (two-way analyses of variance).

**Figure 5 gels-08-00312-f005:**
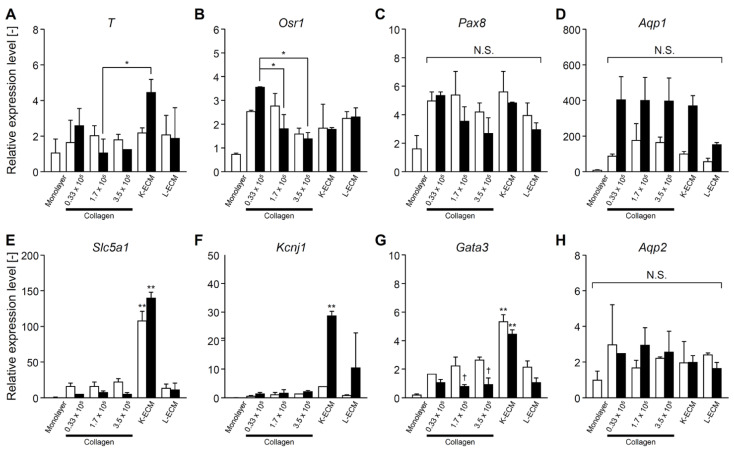
Gene expressions of CHK-Q cysts under several cell density (0.33, 1.7, or 3.5 × 10^5^ cells/cm^2^) and extracellular matrix component conditions. (**A**) *T*, (**B**) *Osr1*, (**C**) *Pax8*, (**D**) *Aqp1*, (**E**) *Slc5a1*, (**F**) *Kcnj1*, (**G**) *Gata3*, (**H**) *Aqp2* (*n* ≥ 2 from at least 2 independent cell preparations). Open and closed columns represent 3 and 7 days of culture, respectively. Data presented as the mean ± SD, ** *p* < 0.01 and * *p* < 0.05 (compared with culture conditions within the same culture time), and † *p* < 0.05 (compared within the culture time under the same culture conditions) (two-way analyses of variance). N.S., not significant.

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
