# Peer review of "Promotion of Cyst Formation from a Renal Stem Cell Line Using Organ-Specific Extracellular Matrix Gel Format Culture System"

_gels, 2022, doi:10.3390/gels8050312_

Round 1
Reviewer 1 Report
In this paper, the authors have successfully established a renal cyst model by using a K-ECM gel. And through a series of in vitro tests, they proved the safety and efficacy of the K-ECM gel. Overall, this is a good research article. However, the manuscript falls short of the required novelty and its have some questions. The detailed reasons are as follows:
- The authors developed a system for culturing stem cells. I don’t think it should be called a gel, it would be more appropriate to describe it as a culture system.
- The authors’description of the results in Figures 2 and 5 is not detailed enough. For example, which genes have increased expression, and what changes compared to the control group.
- The lack of in vivo test results in the research content is not enough to show that the system or gel can provide a good reference for medical applications. If necessary, please supplement in vivo test.
- What are the main advantages of this model established by the author compared to other existing models?
Reviewer 2 Report
The manuscript is clear and relevant for the field, being presented in a well-structured manner. The experimental design is appropriate to test the hypothesis. The manuscript results are reproducible based on the details provided in the methods section and supplementary materials. The figures and schemes/images are appropriate and show the data properly, being easy to understand. The conclusions are consistent with the evidence and arguments presented. The cited references are relevant publications and do not include self-citations.
The authors should provide explanation related to the culture of CHK-Q cells under collagen gel conditions. Is the culture plate also collagen-coated? (Figure 1 A)
Line 73 (Figure 1) “phallidin” should be replated by “phalloidin”.
Lines 108-109 (Figure 3) “phallidin” should be replaced by “phalloidin”.
Line 120 (Figure 4) “phallidin” should be replaced by “phalloidin”.
Line 235: “90 mm colagen-coated dishes” should be completed with “90 mm diameter collagen-coated dishes”.
Line 241: “The 2-passaged cell suspension” should be replaced with “Cells at passage 2”.
Round 2
Reviewer 1 Report
The author carefully revised it as required and suggested to accept it